# Financial Benefits of Mangroves for Surge Prone High-Value Areas

**Henk Jan Verhagen** 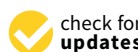

Department of Hydraulic Engineering, Delft University of Technology, PO Box 5048, NL2600GA Delft, The Netherlands; H.J.Verhagen@tudelft.nl

**Abstract:** In this paper, protection options for a high-value, industrial area along the coast of West Bengal (India) are investigated. The options are designed to protect against cyclone surges with a probability of 1/100 per year. Two alternatives are compared, a classical solution of a dike with a revetment and a solution with a mangrove belt in front of the dike. The results reveal that from a pure infrastructural cash-flow point-of-view, the mangrove solution is at least 25% cheaper than the classical solution with a rock revetment. An important finding is that this conclusion does not need the financial evaluation of the obvious additional ecological advantages that mangroves offer. It is postulated that these results are generally valid for high-value coastal areas under the attack of waves during storm surges.

**Keywords:** mangrove; sea defence structure; levee; dike; overtopping; storm surge; cyclone surge

---

## 1. Introduction

The ecological benefits, as well as the many other ecosystem services of mangroves are well known, as has been described in many publications, e.g., in [1]. However, these benefits are partly "virtual" like most ecosystem services [2], i.e., one cannot use these benefits as a financial source for investments in the area. However, mangrove forests also have direct financial benefits, which can lower the construction costs of coastal structures. In this way, nature-based solutions can be very economically attractive solutions.

In the case of a coastal flooding risk, the standard response of coastal managers is to build a dike with a revetment from rocks or concrete blocks. However, in the case of a high value, surge prone area, a nature-based solution may often be cheaper, more flexible (and consequently less risky), and also better for the environment. In [3], this has been elaborated for a coastal dike in an area in Vietnam with intensive agricultural use. In this paper it will be elaborated for a coast along a high-value area in a cyclone prone area. This concrete example is evaluated using real costs (in this case Indian Rupees) rather than making only vague statements like "costs are significantly reduced". The demonstration location selected is the coast near Haldia (see Figure 1). Although concretely evaluated for the Port of Haldia, India, the method described is postulated to be rather universal for cyclone protection against wave attack under storm surge conditions, since the evaluation could be based on cross-shore consideration of hydraulic processes alone without any three-dimensional complexity.

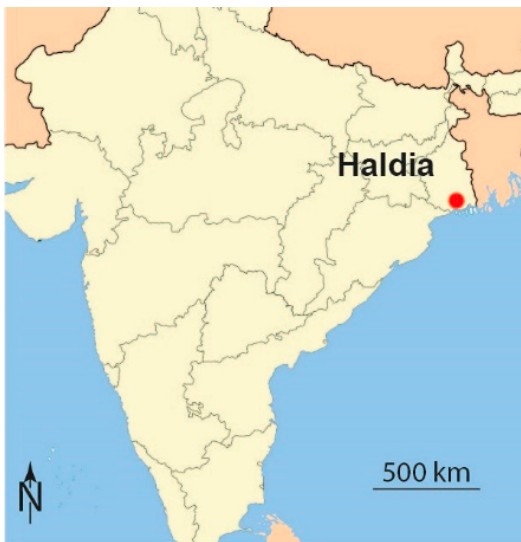

**Figure 1.** Location of the Port of Haldia (map source Wikimedia).

## 2. Materials and Methods

This paper demonstrates that constructing a mangrove forest in front of a revetment will decrease the construction costs of a revetment by a larger amount than the planting and maintenance costs of the mangrove forest. For this purpose, a design is made for a typical revetment in the northeast of India, just north of the Port of Haldia. At this location, there is a risk of cyclone induced flooding, but until now there was hardly any protection against a cyclone surge. Recently, the cyclone Fani (2019) showed the urgent need for good protection. As a method, a proper design was made using classical coastal engineering techniques and compared to the effectivity and costs of an alternative nature-based design. Public data and open source simple models, as well as standard engineering formulas were used.

First, the hydraulic boundary data at deep water for this stretch of coast was determined (i.e., water levels and wave data). Second, the waves at the border of the shallow zone and at the toe of the dike were derived. This was done with and without a mangrove forest. Third, a sea dike was designed to protect the land against the cyclone surge. Finally, the cost of both alternatives was determined and compared.

For all steps, standard handbook-engineering methods were applied, and freely available software was used. It was demonstrated that for the basic decision whether a mangrove forest is economically interesting, there is no need for advanced modelling tools. For making the final design of the protection, some more advanced calculations are recommended for fine-tuning the design.

## 3. Computations

### 3.1. Case Study Location

For this study, a coastal stretch in India was selected, in the state of West Bengal, just north of the Port of Haldia. More precisely, along the India Exchange Place Road (22°3′59′′ N; 88°9′24′′ E), see Figures 2 and 3. At this specific location, a sea dike was constructed at a location where a mangrove belt had disappeared.

At the site there is a larger landfill; the fill area to the west of the site was used to construct some chemical plants. The area at the demo site is still vacant. The vacant lot is separated from the river by a road, the India Exchange Place Road. Seaward of the road there is a 25 m wide zone with sparse mangrove vegetation, fringed by a mudflat.

For industrial development, jetties are being constructed. At the tip of the jetty some dredging is needed to make the jetty accessible for ships, the natural depth of the Hooghly River at this point is only 0.5 m below chart datum [4,5].

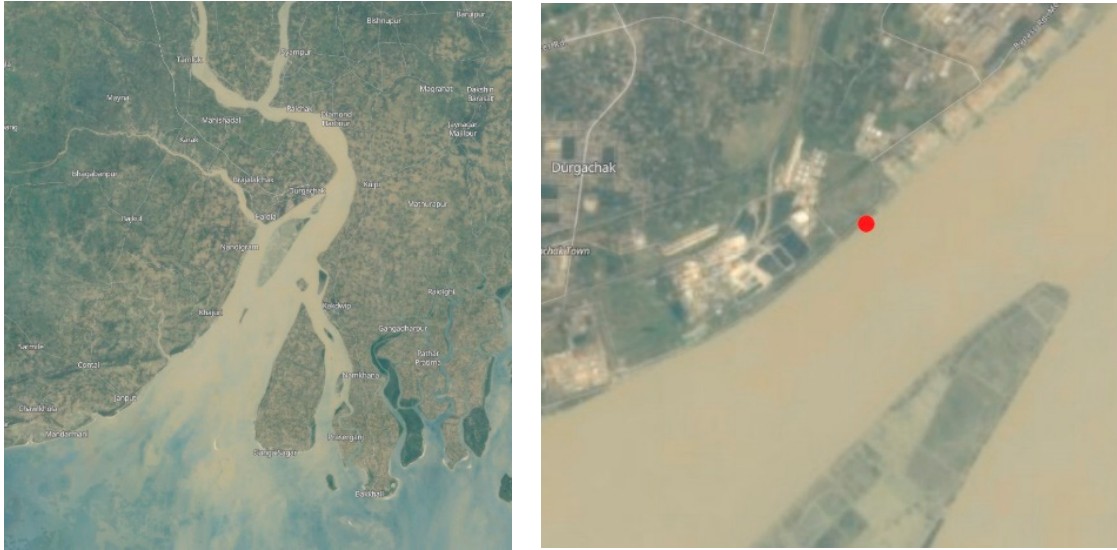

**Figure 2.** Location of the demonstration area, northeast of Haldia port, (an overview (**on the left**) and a more detailed view (**on the right**), © OpenStreetMap contributors).

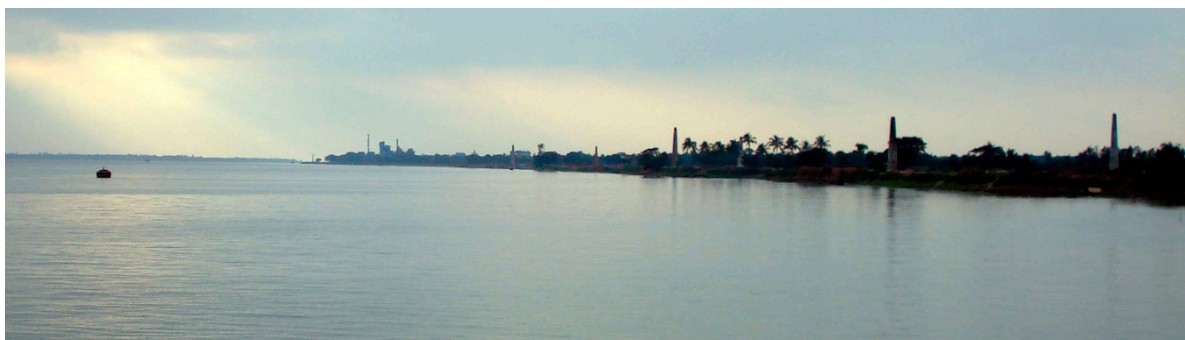

**Figure 3.** Overview of the coastline at the demo site (Photo by Shiv S Tripathi, Wikimedia Commons).

*3.2. Hydraulic Boundary Conditions*

At this location, both waves and water levels are relevant. One of the problems in this area is flooding due to cyclone surges. It is assumed that the dike has to protect this area against a 1/100 year surge condition. The normal tide at this location is considerable. The port of Kolkata even publishes warnings for the very extreme tides, called "tidal bores" in a yearly publication [6]. These "tidal bores" occur approximately 14 times per year. Tide levels are:

Neap tide: Low water 1.8 m–High water 4.5 m

Spring tide: Low water 1.0 m–High water 6.5 m

All figures are above Chart Datum (see Figure 4). The mean sea level (MSL) at this site is 3.5 m above Chart Datum. Data are based on tidal observations in June/August 1991 at the Oil Jetty just west of the demo site, and are given in [7].

During cyclone events, the water level will rise considerably. In 2013, Rao [8] presented a prediction for the coast of India for the present sea level and for a rising sea level. In the calculation in this paper, sea-level rise is not included. According to Rao, the present 1/100 year water level in West Bengal is 9.6 m above Chart Datum. This is confirmed by [9]. Because of the funnel-effect of the Hooghly River, for this computational example, a somewhat higher water level was used: 10 m above Chart Datum.

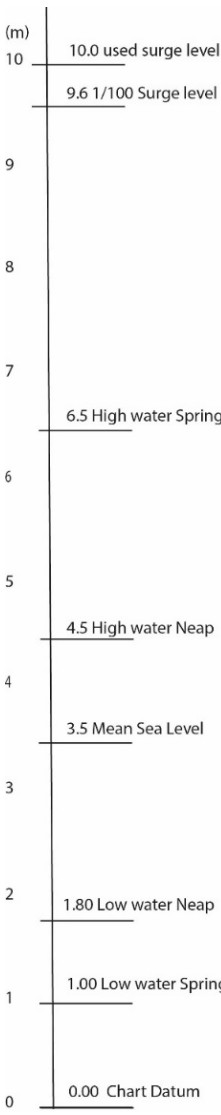

**Figure 4.** Water levels.

According to [10,11], wave heights during cyclones in the Bay of Bengal can reach up to 9 m with periods of 12 s. There are no accurate data on the return periods of wave heights, so the deep water wave height is not really known. Fortunately, as will be seen later, this is also not really needed. Feeding these data into the computer program Swan showed that waves break because of shallowness. Computations with wave heights of 7, 9, and 11 m on deep water give very similar results in the channel in front of the coastline. A computation was made for a water level of 6 m and 10 m above Chart Datum. This also showed that the influence of the water level on the resulting wave height near the shore is much larger than the influence of the deep water wave height.

The height of the landfill behind the India Exchange Place Road is not known but based on photographs it is estimated as a little above High Water Spring (HHWS), see Figure 4; therefore, the height of the fill is assumed to be 7 m above Chart Datum. This means that during a 1/100 year cyclone, the industrial area will be flooded. Such flooding causes direct damage to the installations, but also causes follow-up damage because there is some time when production comes to a standstill. Moreover, flooding of the (chemical) installations will cause considerable environmental damage.

To prevent such flooding, a sea defence structure can be built (a sea dike). In this paper, the cost of a classic sea dike is compared with a sea dike with a mangrove belt in front of it.

### 3.3. Wave Calculation without Mangroves

In order to determine the design wave height in front of the mangroves and/or revetment, a calculation was made. Under normal conditions there is hardly any wave action at this site, because it is very shallow and protected by the Char in front of the site. The computations were carried out with the computer program Swan. Because two-dimensional effects are not very relevant for this problem, the computation was done with SwanOne [12]. In fact, SwanOne is a simplified user interface for Swan. In order to make wave computations, a cross-sectional profile was needed. Figure 5 shows the used profile, which is based on data from Navionics [5].

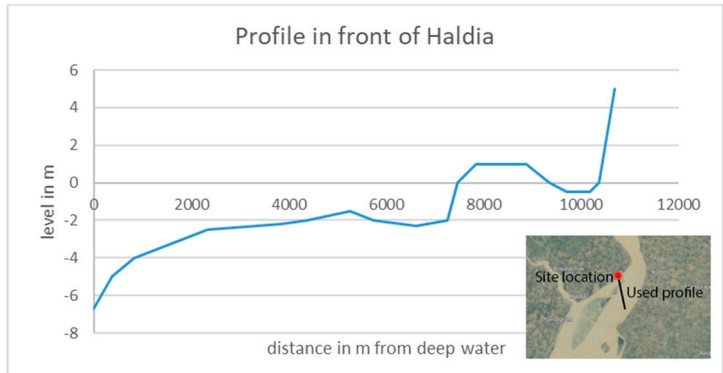

**Figure 5.** Profile in front of the coast of Haldia (data from Navionics [5]).

The starting point for the calculations is located south of Haldia, already inside the Estuary. As mentioned before, wave data at that location are not exactly known, but it is assumed that they are not much lower than in the Bay of Bengal. Figure 6 shows the results of the computation during normal high water (left the full profile, right the last few hundred meters).

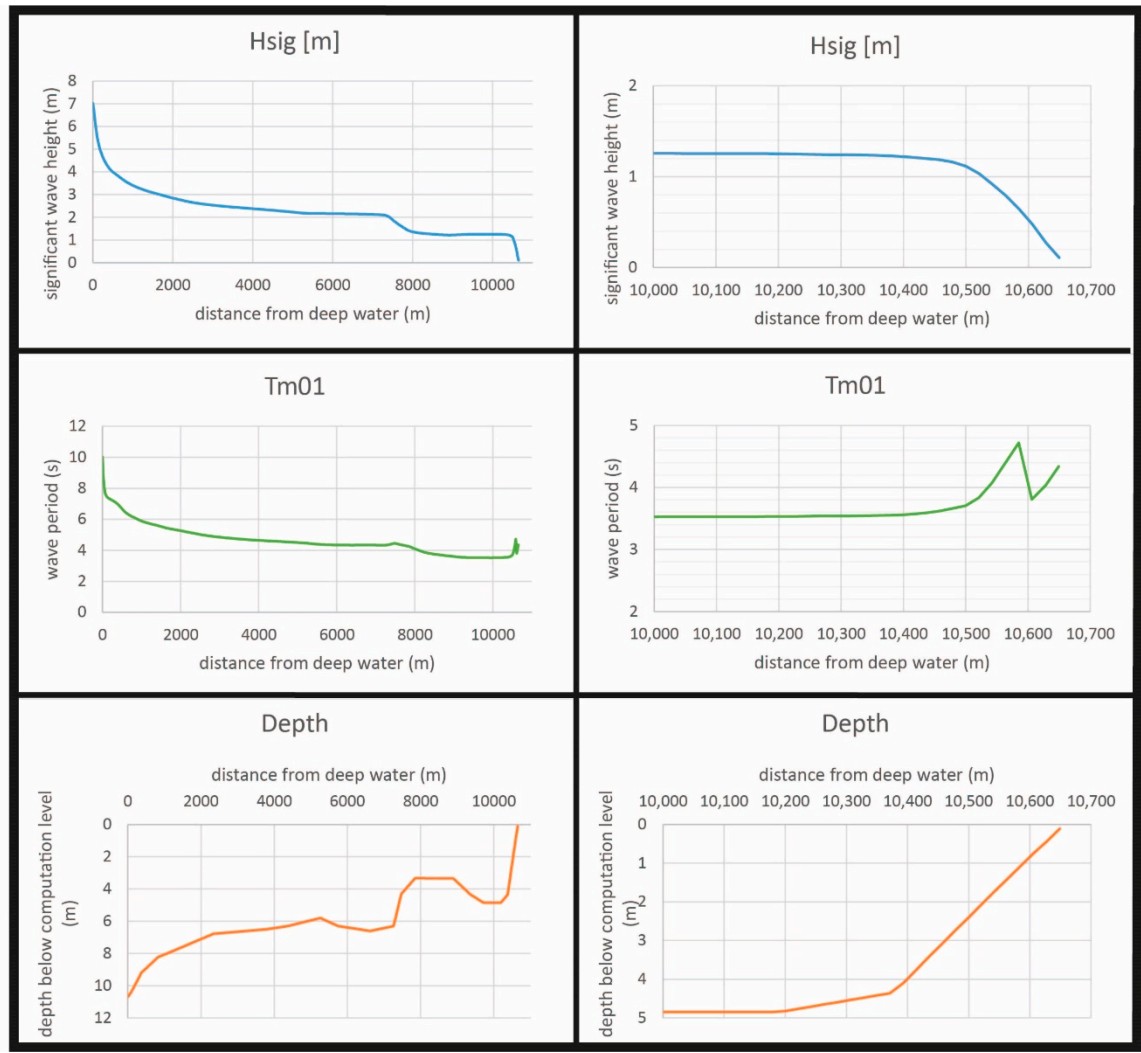

**Figure 6.** Wave height computation for normal high water (Chart Datum +6 m).

As can be seen, the wave height decreases fast because of the shallowness of the water. Waves will just pass the Char, but will be broken. At a water depth of 3 m (this is 3 m above Chart Datum), the wave height is 1.2 m and the wave period is reduced to somewhat more than 4 s. Figure 7 shows that the original deep water wave height is not very influential, near the coastline the difference is only in the order of 10 cm.

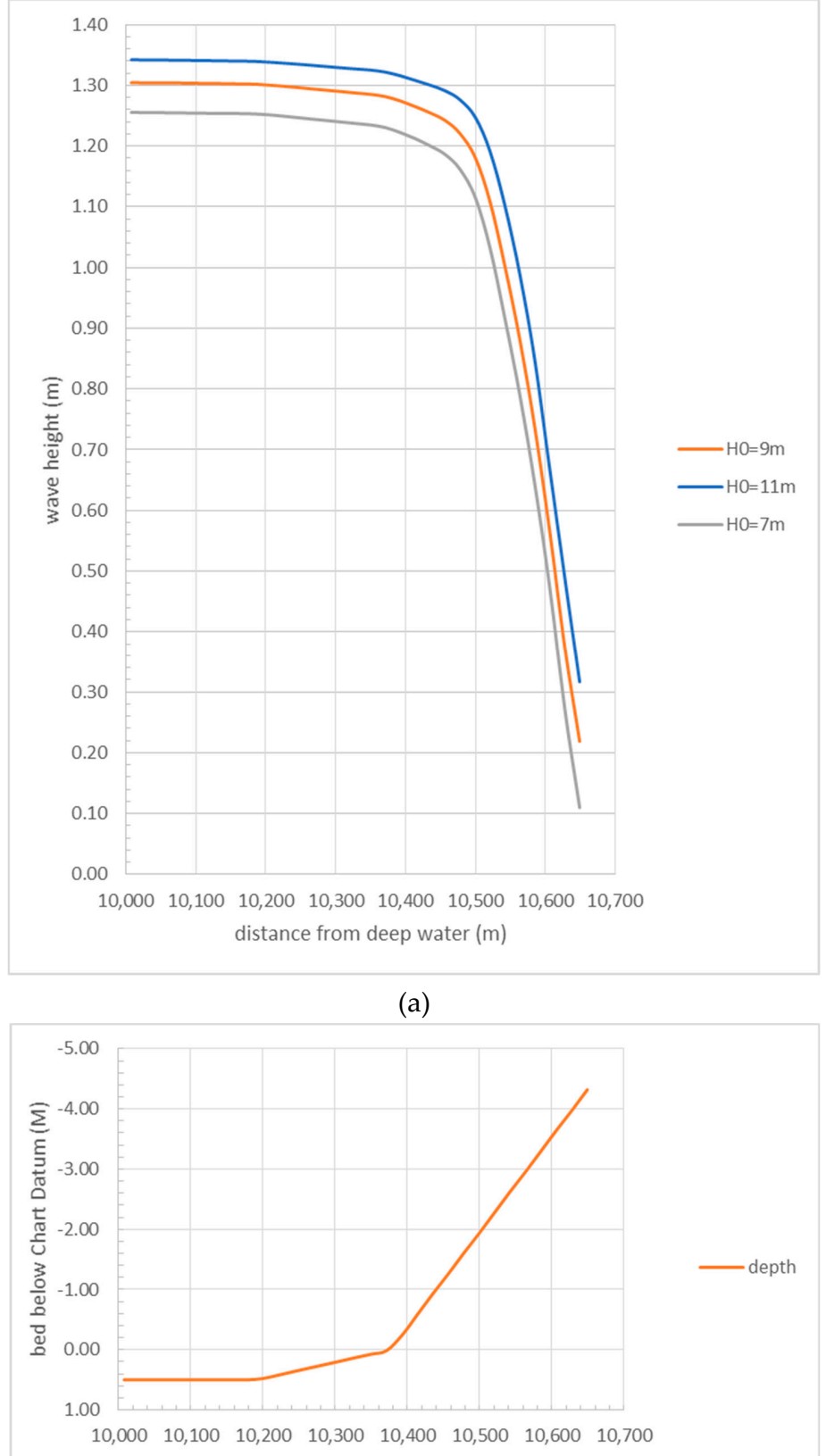

(a)

(b)

**Figure 7.** Wave height computation for three different wave heights (**a**) and the water depth at the same location (**b**).

A similar computation can be carried out for the situation with a cyclone surge. Then the water level is 10 m above Chart Datum, see Figure 8.

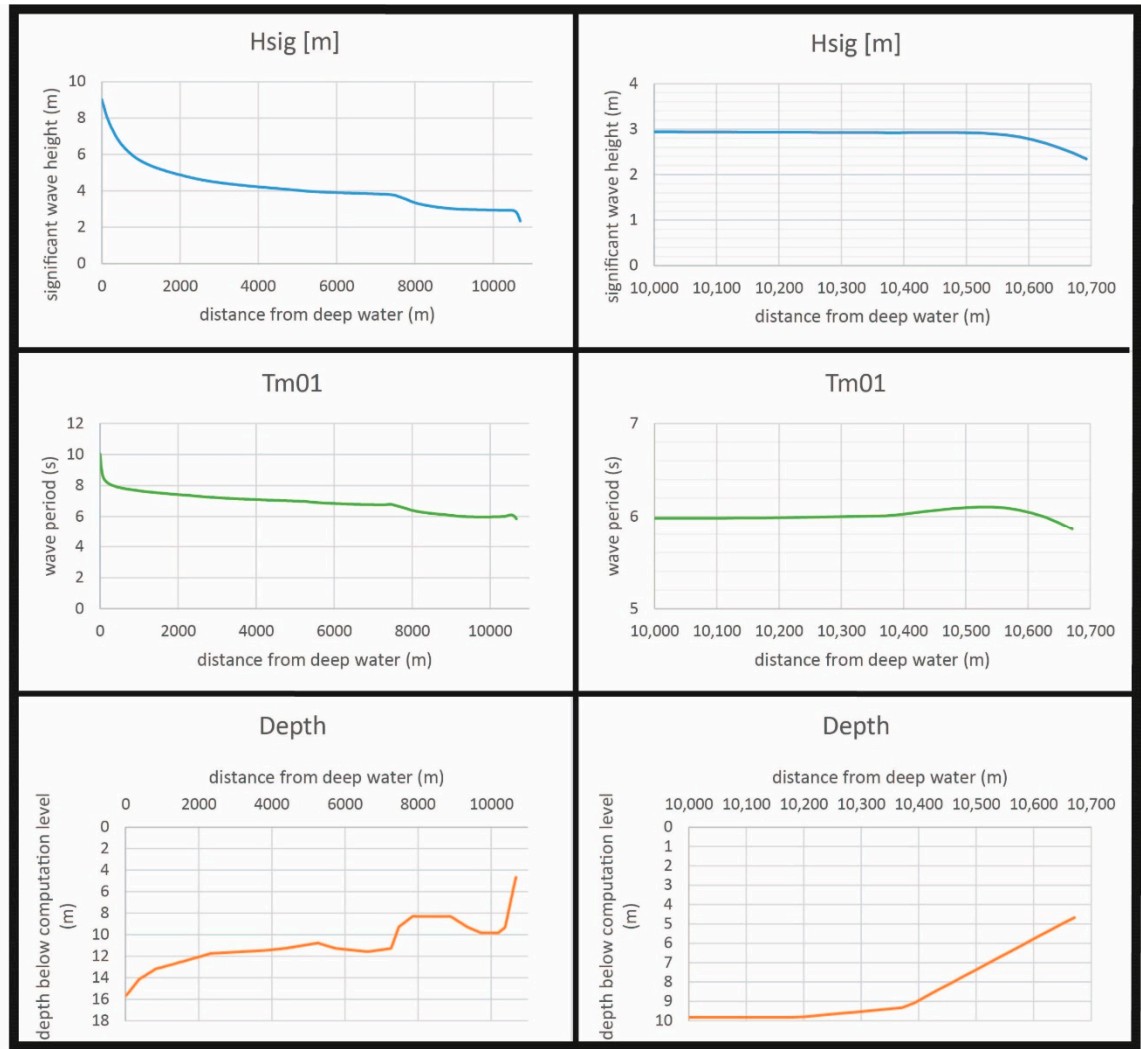

**Figure 8.** Wave height computation for a 1/100 year cyclone surge (Chart Datum +9 m).

At the same shore point (10,600 m from deep water) the wave height is now 3 m, and the period 6 s. This implies that a shore protection at this location has to be designed for a surge level of 10 m above Chart Datum, a significant wave height of 3 m and a period of 6 s.

However, this period is rather misleading. The wave energy spectrum at that location reveals that the 12 s component is still quite prominent, and contains a lot of energy. This is relevant because the overtopping by waves over a sea defence mainly depends on the longer waves in the spectrum. For this reason, the parameter $T_{m-1,0}$ (The parameter $T_{m-1,0}$ is the period calculated from the first negative moment of the energy spectrum) was used. This value is for the case with 10 m surge $T_{m-1,0} = 8$ s. For the 4 m level it is $T_{m-1,0} = 7$ s (Figure 9).

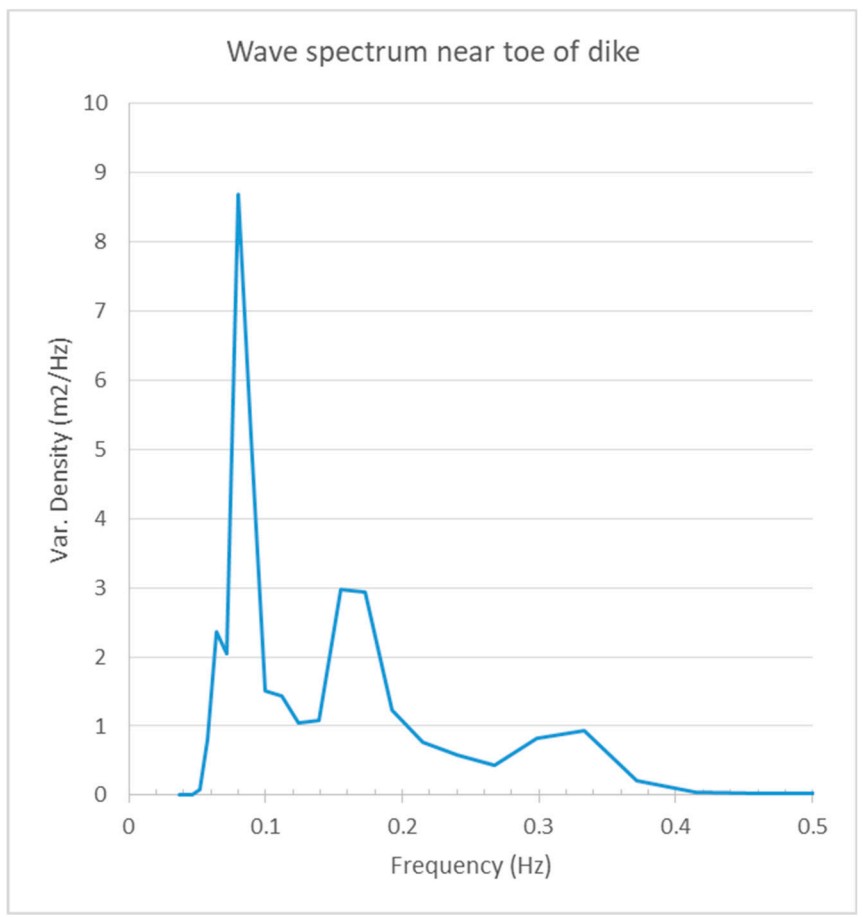

**Figure 9.** Wave height computation for a 1/100 year cyclone surge of 10 above Chart Datum.

### 3.4. Wave Calculation with Mangroves

SWAN offers the possibility to include dissipation by mangroves in the computation. However, that is complicated and requires quite some effort. For this paper, being only interested in the overall effect, simple graphs can be used, as has been demonstrated in [3]. Figure 10 gives an example.

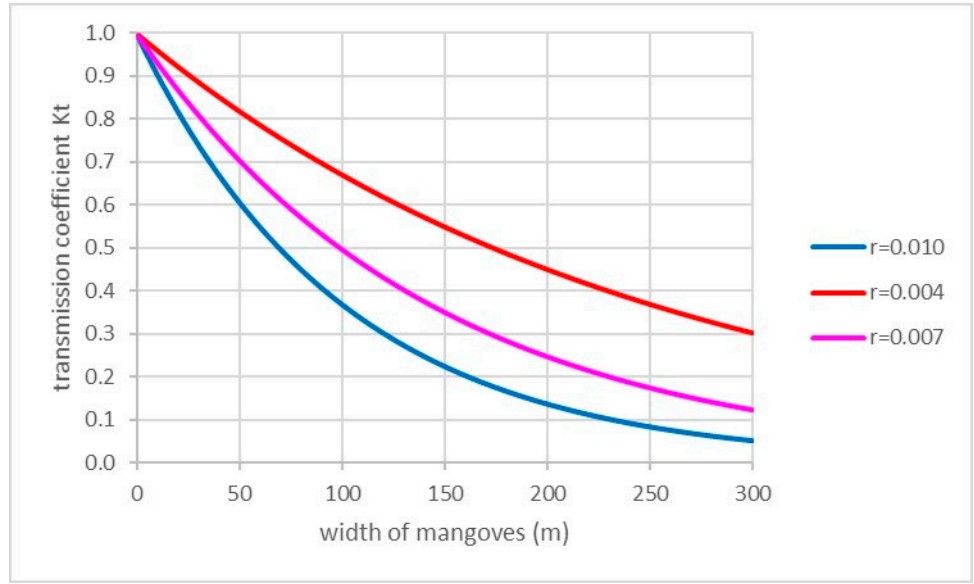

**Figure 10.** Dissipation coefficient $K_t$ as a function of mangrove width (data from [2]).

The value $r$ is an indication of the mangrove density ($r$ = 0.010→dense vegetation, $r$ = 0.007→average vegetation, $r$ = 0.004→sparse vegetation). The value of $r$ in a real mangrove area is highly variable. This has been discussed in [3], it was found that the Quynh formula gives quite good results by using easily obtainable data. The Quynh formula [4] is given as Equation (1):

$$b = -0.0481 + 0.0016H_{vn} + 0.0017\ln(N) + 0.0077\ln(Tc) \tag{1}$$

where:

$H_{vn}$  average total height of the trees in the forest (m)
$N$     density of mangrove trees taller than 1 m (trees/ha)
$T_c$    Forest canopy cover (%).

Using the Quynh formula with a tree height of 6 m, a density of 150 trees per ha, and a canopy coverage of 75%, gives a value of $r$ = 0.004.

So in order to decrease the wave height from 3 m to 1 m, one needs 150 m of mangroves with average density. These values will be used to calculate the required width of a mangrove belt in order to decrease investments in dikes and revetments. Of course, one has to take into consideration that mangroves only grow abundantly between MSL and HHWS (so in this case between Chart Datum + 3.5 m and Chart Datum + 7.0 m). The value $K_t$ is the transmission coefficient, i.e., the reduction of the wave height due to the mangroves. The relation can be written as [3]:

$$H_t = H_0 K_t = H_0 e^{-rx} \tag{2}$$

in which:

$H_t$    resulting (transmitted) wave height
$H_0$   original wave height (at the beginning of the mangrove belt)
$K_t$    transmission coefficient
$r$     reduction coefficient.

This equation can be used to adapt the found results from SwanOne.

Figure 11 shows the calculated wave heights near the coast without mangroves (red line) and with mangroves (grey line, using the coefficient for sparse vegetation). It can be seen that during a cyclone surge of 10 m, the wave height decreases from 189 cm to only 71 cm. This reduction is relevant, because for a 1:3 dike, good quality grass cover can withstand these waves, while for waves of $H_s$ = 1.89 m, a rock revetment is needed (with rather heavy rocks). In this example calculation, a rather pessimistic mangrove density has been applied ($r$ = 0.004), because, especially in the beginning, the mangrove belt will not be very dense. The total width of the belt is approximately 250 m. More is not possible, because mangroves will only grow between MSL and HHWS.

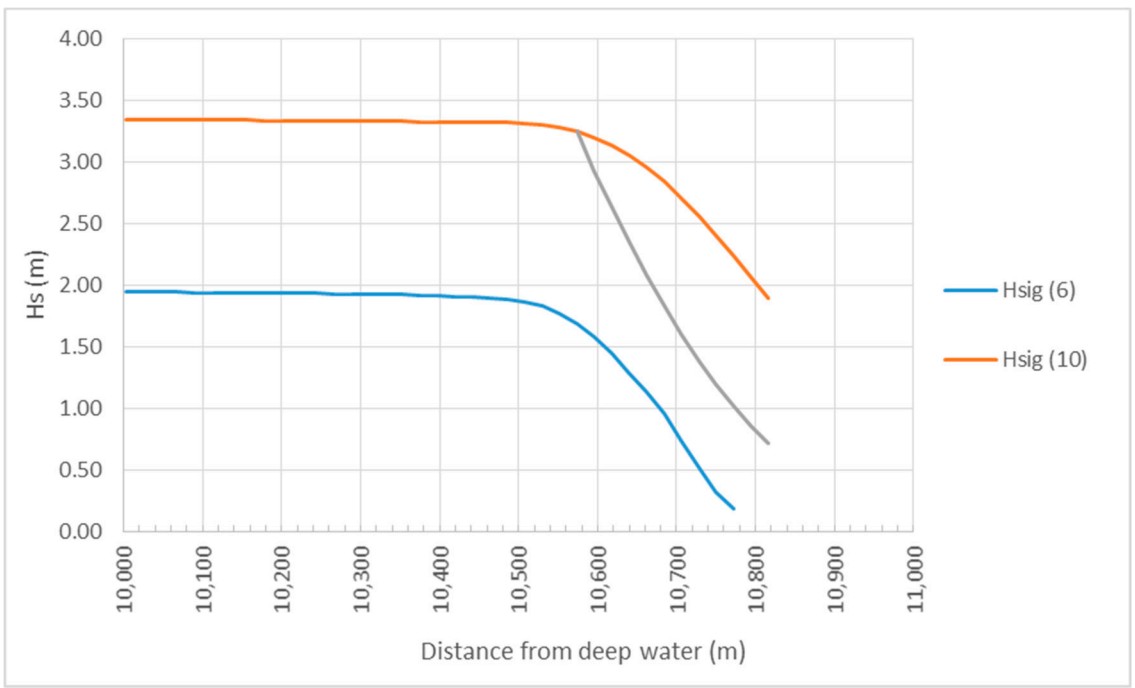

**Figure 11.** Wave height near the coastline for a surge of 6 and 10 m.

### 3.5. Design of a Dike without Mangroves

The sea defence dike has to be designed in such a way that during design conditions, the amount of overtopping is limited. When there is too much overtopping, the inner slope will start to erode, which eventually will lead to a breach in the dike. For classical design of dikes, a dike is sufficiently safe when less than 2% of the waves will overtop the dike. The computations were carried out according to the EurOtop Manual [13]. A slope (both outer and inner slope) of 1:3 was applied, because experience has shown that with such rather gentle slopes, one will not have problems with geotechnical stability. More gentle slopes usually give a higher ecological benefit but are more costly because of their land use. The height of the dike can be reduced by constructing a berm on design water level (in this case the 10 m cyclone level). A computation is made for a case without a berm (dike 1) and a case with a 10 m berm (dike 2). The results show that the dike with berm is lower and has a smaller volume than the dike without a berm. Usually there is a road on top of the dike, but the cost and effects of such a road are not included in this analysis, because they are the same for all options analysed. The result is shown in Figure 12 (note that in Figures 12 and 13 for the vertical axis depth is positive, so values above Chart Datum are negative). The actual computations can be found in the spreadsheet in the Supplementary Material.

As mentioned earlier, for an attack by waves with a height of 1.89 m, a dike needs to be protected with rock or revetment blocks. In this case, a rock revetment was selected, because that is more common in India. A placed block revetment will have approximately the same price. The required rock size was calculated with the Van der Meer formula with a Notional Permeability coefficient P = 0.1 (because the dike is impermeable for waves). The calculation was done according the Rock Manual [14]. This results in rock with a weight of 2 tonnes. The layer thickness (a two stone layer) was 1.2 m. This protection has to stretch from the toe to the crest of the dike.

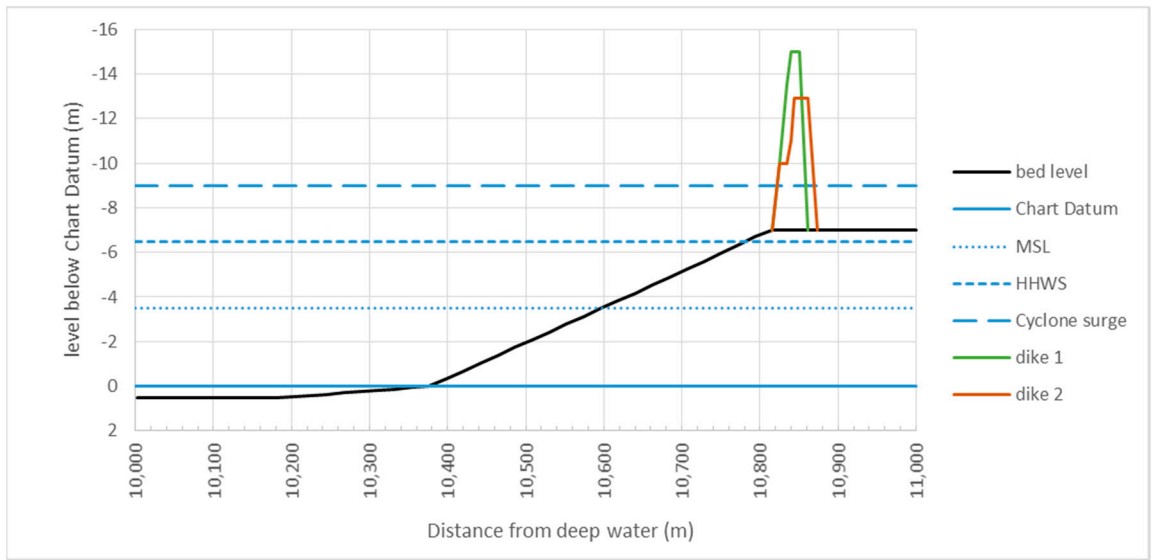

**Figure 12.** Geometry of the dike without mangrove belt.

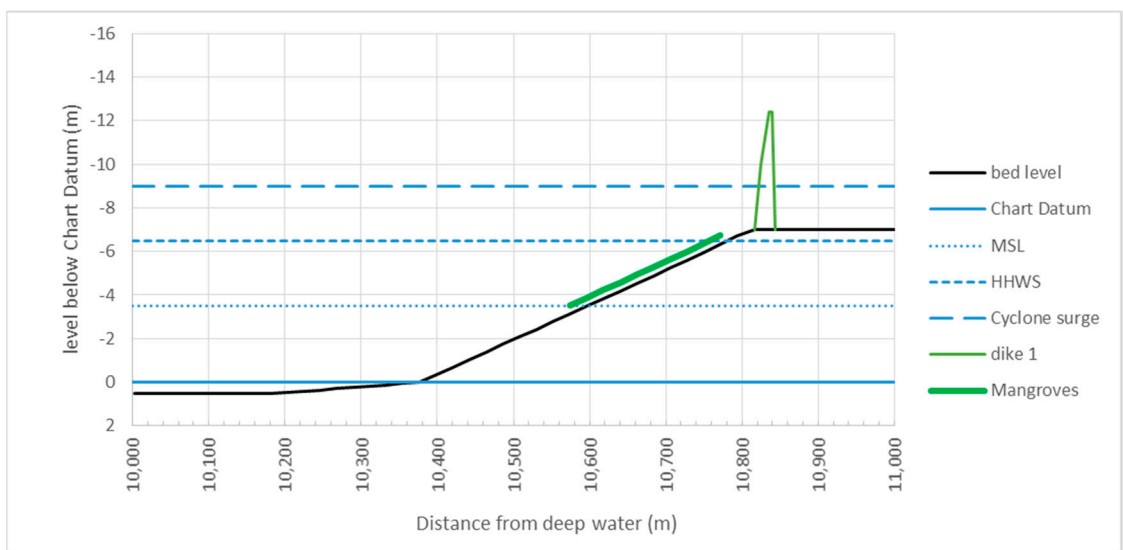

**Figure 13.** Geometry of the dike with mangrove belt.

*3.6. Design of a Dike with Mangroves*

For a dike with a mangrove forest in front, the same calculation can be done. However, now the wave conditions are much milder. This results in a lower dike with a much smaller volume. The result is shown in Figure 13.

The dike now reaches 12.4 m above Chart Datum (i.e., 3.4 m above the landfill), while the dike without mangroves reached 6.0 and 3.9 m above the landfill. So this dike is not much lower than the dike with berm in the first case, but is much narrower. Also, no rock revetment is needed, which is a very important benefit. Constructing a berm in this dike does not give a significant reduction of the volume, therefore it was not considered. The actual computations can be found in the spreadsheet in the Supplementary Material.

## 4. Costs

### 4.1. Cost of the Dike without Mangroves

For the calculation of costs, one needs the price of materials. In order to get a first impression, very global figures are used. The costs are expressed in Indian Rupees (INR). In mid-2019, the exchange rate was 1 INR = 0.0131 Euro. Unit prices were based on values found on the internet, but mainly on the data presented in [15], which is based on [16]. The following unit prices (including transport and handling) were used:

- Soil for the construction of the dike: 500 INR/m$^3$
- Purchase of land for dike construction: 20,000 INR/m$^2$
- Price of rock, including handling, filter, etc.: 1000 INR/tonne

This results in the costs per running meter of dike as given in Table 1:

**Table 1.** Cost of dikes without mangroves.

| Values in Indian Rupees (INR) Per m Dike | Dike 1 | Dike 2 |
|---|---|---|
| Purchase of land | 1,160,000 | 1,100,000 |
| Construction of the dike body | 353,500 | 185,500 |
| Construction of the revetment | 52,992 | 39,082 |
| Total cost | 1,566,492 | 1,324,582 |

### 4.2. Cost of the Dike with Mangroves

A similar calculation can be made for the dike behind the mangrove belt. Because the dike is smaller and there is no revetment needed, this will be considerably cheaper. However, a mangrove belt has to be created with a width of 250 m. The cost of the mangrove seedlings can be neglected, and planting is not very expensive. Experience in Vietnam [17,18] has shown that planting mangrove seedlings can be done with a speed of 1–5 m$^2$/h, depending on the degree of muddiness of the area. For this example, we used 2.5 m$^2$/h. So 1 m width of mangrove-planting for the full 250 m of the foreshore will take some 100 h. Given the average wages of land workers in India, this will cost in the order of 6000 INR per meter of dike.

Because seedlings cannot survive significant wave action, a temporary wave barrier has to be constructed (see example in Figure 14). This can be a simple bamboo screen, as is used in many places of the world. Such a screen can be built by a crew of five men, they make approximately 10 m per day, so including the materials this will cost in the order of INR 1000 per running meter of dike.

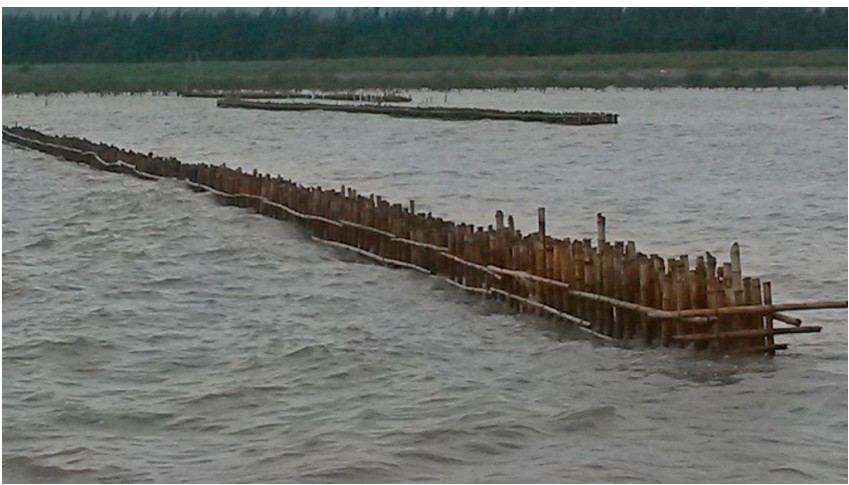

**Figure 14.** Bamboo with reed filling as temporary protection for mangrove planting in Vietnam (picture by author).

Unfortunately, there is a risk of damage to the mangrove plantation in the first years due to cyclones. If there is a significant cyclone during the first 5 years of the project, there is a risk that the mangroves are washed away and have to be replanted. Let us assume that the probability that this happens is in the order of 50%, and that the wave protection screen has to be rebuilt two times during this first 5 years. This will increase the planting cost by 50% and the screen cost by a factor of 3. In total, this will give the cost as presented in Table 2.

**Table 2.** Cost of dikes with mangroves.

| Values in INR Per m Dike | Dike with Mangrove |
| --- | --- |
| Purchase of land | 860,000 |
| Construction of the dike body | 138,500 |
| Mangrove system | 7000 |
| Reservation for cyclone risk | 5500 |
| Total cost | 1,010,900 |

Comparing Tables 1 and 2 leads to the conclusion that a 1/100 flood defence for a 1/100 year cyclone event (with level of 10 m above Chart Datum) based on mangroves and a small dike costs only 75% of a similar flood defence based on only a dike and a rock revetment.

In this specific example (for an industrial estate), the price of land is the determining factor. In this case it is 85% of the total cost. A high price of industrial land is realistic, but even if the value of the land is only 10% of the value assumed, the main conclusion is still valid. In percentage, the savings are even larger. When using a price for purchasing the land of only 2000 INR/m$^2$ the cost of dike 1 becomes 521,000 INR/m, and dike 2 becomes 336,000 INR/m, while the case with mangroves will cost 237,000 INR/m. So in this case, the dike with mangroves is 55% and 29% cheaper than dike 1 and dike 2, respectively.

In some cases, land is already government owned and does not have to be purchased for the construction of a revetment. However, even if the land does not need to be purchased, the land has a value. For example, in Vietnam, where market values of (rural) land hardly exist, the land certainly has a value. Therefore, for the Vietnam case [3], the "market" value of the land was not used, and instead the cumulative yield of the rice produced on that land was used.

This means that, besides the non-monetary benefits of mangroves like ecosystem services, the application of mangroves in coastal flood protection gives a cash benefit. Figure 15 shows a typical example of local mangroves which can be used for the creation of a mangrove belt.

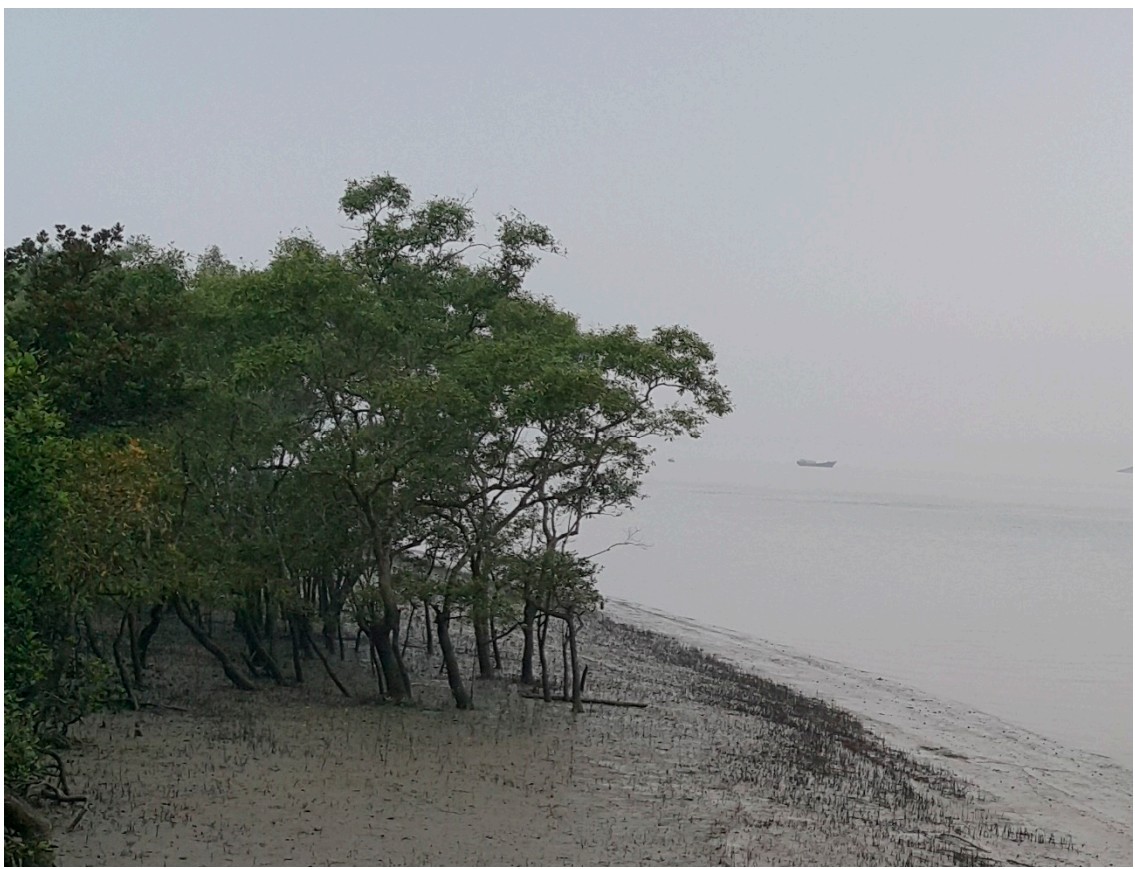

**Figure 15.** Photo of mangroves in the area (Photo by Hsinha, Wikimedia Commons).

## 5. Discussion

Comparing various images from Google Earth showed that at this specific location, there is no bank erosion in the long run. This is quite fortunate for the development of a mangrove belt. In the case of eroding coastlines, the creation of a mangrove belt can become quite a challenge. Usually the erosion mainly occurs below the MSL-line, while the mangroves are above that point. The consequence is then that just seaward of the MSL-line there will be a significant steepening of the bed. Such a steepening eventually leads to failure of the slope and the seaward edge of the mangrove will fall into the water. Mangroves will not stop this type of erosion. Only erosion caused by perpendicular approaching waves will be stopped by mangroves, because they decrease the wave height at the point where this type of erosion occurs. This is the case when the erosion occurs during a cyclone surge. When the waves enter obliquely, and the erosion is caused by a gradient in the longshore transport, mangroves hardly have any effect on the erosion. Basically, the erosion also occurs below the MSL-line, where there are no mangroves. Also, erosion by strong river currents (especially in the outer bends of a river) are not stopped by mangroves.

From this analysis, one can conclude that mangroves can be a cost-effective contribution to the protection of low-lying lands against cyclone surges. This is a direct cash benefit, because the construction is simply less costly. The calculation in the example above is rather simplified. Therefore is should not be seen as a basis for a design at this specific location. For such a design the hydraulic boundary conditions should be determined with much more accuracy. This applies both to the "dike only" solutions as well as to the "mangrove" solution. The saving of approximately 25% of the cost will remain the same.

The economic savings due to a mangrove belt are threefold:

- Because of the lower wave height:

  ○ The revetment can be lighter (and therefore cheaper)
  ○ The crest can be lower

- Because of the lower crest

  ○ The volume of the structure is less (and therefore cheaper)
  ○ Less land has to be purchased (and therefore cheaper)

Additional benefits, apart from the ecological benefits, are that the construction of a mangrove belt is cheaper, but requires much more manpower from unskilled or low-skilled labour. Especially in countries with high unemployment in the lower educated parts of the population, this can be a significant advantage. When the mangrove belt is constructed by local labour, the labourers also have the feeling that the belt is "their own", and are probably more reluctant to damage it (e.g., for gathering firewood). The rock revetment is always made with heavy equipment and rock from elsewhere. This does not strengthen the ownership of the local population.

## 6. Conclusions

This paper demonstrates that the use of mangroves as coastal protection against flooding during cyclones can be beneficial for areas with a relatively high economic value, like industrial estates. Purely based on economic advantages, and even neglecting the additional ecological benefits, a revetment with foreland mangrove protection is less costly than a revetment without such a foreland. In order to quantify these benefits, an example is given for the coast near Haldia, India. Even based on first order calculations, it becomes clear that a mangrove protection is significantly cheaper than a protection without mangroves. It is postulated that these results are generally valid for high-value coastal areas.

**Supplementary Materials:** The following material is available online at http://www.mdpi.com/2073-4441/11/11/2374/s1, Spreadsheet: DikeCalculation.xlsx.

**Funding:** This research received no external funding.

**Acknowledgments:** Figure 2 is based on open data, licensed under the Open Data Commons Open Database License (ODbL) by the OpenStreetMap Foundation (OSMF).

**Conflicts of Interest:** The author declares no conflict of interest.

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
