# Peer review of "Financial Benefits of Mangroves for Surge Prone High-Value Areas"

_water, doi:10.3390/w11112374_

Round 1

Reviewer 1 Report

1. In this study, Authors proposed financial effect of reduction dike crest by using mangroves group plants against cyclone surges without damaging the ecosystem.

2. Dike model test is required, it is must be for reliable verification.

3. I think this idea is very creative, crest of dike can be lowered due to the effect of damping  tidal bores.

4. However, it may be advantageous in the future to make suitable dike  crest for  safety of industrial lands.  

Author Response

Thank you for your analysis and comments. I addressed them below. Also the language will be reviewed by an expert.

In this study, Authors proposed financial effect of reduction dike crest by using mangroves group plants against cyclone surges without damaging the ecosystem.
Indeed, the paper show the financial effect

Dike model test is required, it is must be for reliable verification.
I agree that for final application test are needed, but I very much doubt if model tests will be able to verify this approach. It is nearly impossible to make a very reliable scale model of a mangrove forest. Until now all model tests in the worlds have been done with very schematized “mangroves”. This is due to various scale effects. Lab tests are only possible on large scale. An example are the recent tests in the Deltaflume of Deltares in the Netherlands where full scale vegetation has been tested. See for this their website: http://woodsversuswaves.com/

I think this idea is very creative, crest of dike can be lowered due to the effect of damping tidal bores.
As explained in the paper it is not really the bore which is lowered, as the wind-waves on top of this tidal bore.

However, it may be advantageous in the future to make suitable dike crest for  safety of industrial lands.
Indeed,  vegetation can contribute to lower the cost of sea defence.

The full text will be reviewed on English language and style by an expert.

Reviewer 2 Report

The paper describes the comparison of cost for the revetment and mangrove protection belt in front of a dike. The wave modeling was carried out in ONE-SWAN model and the mangrove belt is expressed in the permeable layer with transmission coefficient. The simplicity of the modelling makes the paper easy to be read but the calculated cost is not so simple. One of the discussions is on the reason of applicability of detail cost calculation. The additional comments are as follows;

1: Please make “Conclusions”

2: Please indicate the location of “Haldia”Port in the Indian map.

3: Line 67; The expression of tide is very complicated. If possible, make a figure indication the level of tide on a vertical pole.

4: Figure 4: The upper two figures are the same. Why the period starts from 10second in the left middle figure?

5: Figure 8: The mangrove density is not numerically determined. How do you express the density by “dense”, “average” and “sparse”?

6: Figure 9: In the figure, what dense mangrove is applied for the gray calculation curve?

7: Table 1; The calculation cost is too detail. Why can the revetment cost be estimated by the unit one?

8: Line 236: What does the 1/100 flood level correspond to?

9; Please enter a photograph of local mangrove.

Author Response

Thank for your remarks and noticing omissions in the paper. Below I have addressed your points in more detail. The final text will be reviewed by an expert in English language.

Please make “Conclusions”
A section “conclusions” has been added. I added a copy of this paragraph below this text. Please indicate the location of “Haldia” Port in the Indian map.
A small map indicating Haldia is included. Line 67; The expression of tide is very complicated. If possible, make a figure indication the level of tide on a vertical pole.
A figure with the levels is added. Figure 4: The upper two figures are the same. Why the period starts from 10second in the left middle figure?
Indeed the upper left figure is a copy of the upper right one. This is an incorrect figure, has been changed. Swan immediately adapts (physically) incorrect input to real values, the used input at the deep water boundary cannot exist there (because of breaking due to limited depth) and therefore in Swan the wave breaks and both Hs and Tm01 go down very soon. Figure 8: The mangrove density is not numerically determined. How do you express the density by “dense”, “average” and “sparse”?
This is explained in detail in the publication I refer to in this paper (Verhagen 2018), but it is indeed convenient to repeat this in this paper. The paper has been improved at this location, also a formula to calculate r is included in the paper now. Figure 9: In the figure, what dense mangrove is applied for the grey calculation curve?
The coefficient for sparse vegetation is used, this is added to the text. Table 1: The calculation cost is too detail. Why can the revetment cost be estimated by the unit one?
The currency-unit was missing, this has been improved. Line 236: What does the 1/100 flood level correspond to?
This is a level of 10 m + CD, this has been added into the paper. Please enter a photograph of local mangrove.
A photograph is added.

===============================

7. Discussion

In this paper it is demonstrated that the use of mangroves as coastal protection against flooding during cyclones can also be beneficial for areas with a relatively high economic value, like industrial estates. Purely based on economic advantages, and even by neglecting the additional benefits of the ecosystem services, mangrove protection is less costly than making revetments without a foreland consisting of a healthy mangrove belt. In order to quantify these benefits an example is given for a coast near Haldia, India. Even with rough calculations, it becomes clear that a mangrove protection is significantly cheaper than a protection without mangroves.

Reviewer 3 Report

In my opinion, the manuscript entitled “The financial benefits of mangroves for surge prone industrial areas” should not be accepted for publication according to the following comments.

- The article is for a specific case study so the title is not appropriate.

- The specific case study is presented in the form of a rough preliminary study and not as a detailed final design; there are too may assumptions and simplifications.

- In any case, most of the financial benefit results from the value of the land purchase and not the technical works themselves. This aspect is not explained well, and may not be relevant in other countries where one does not have to purchase the land where coastal protection works are constructed.

Author Response

Thanks for your comments.

I have tried to improve the paper according to your comments. However, I prefer to stick to the title I have given to my paper, because I would like to demonstrate that mangrove protections are not only relevant for rural coasts with low value land, but also for land with a high value, like industrial estates.

This article was not intended as a case study for a specific location, but as a demonstration that for coasts along industrial estates in cyclone prone areas mangroves can be an economically viable solution for flood protection (a mangrove belt lowers the cost of the revetment with an amount which is larger than the cost of the mangroves). To demonstrate this, it is more clear to show a concrete example using real costs (in this case Indian Rupees) than only vague statements like “costs are significantly reduced”. However, in order to make an example with real costs, one has to select a demonstration location. In this case as demonstration location the coast near Haldia was selected.

From the remarks of the reviewers it is clear that this approach was not clear, and therefore it is explained more in detail in both the introduction as well as in the (new) section “conclusions”.

At the end of section 3 (between brackets) it was already stated that this study was for demonstration purposes, and should not considered as a final design. In section 6 is mentioned: The calculation in the example above is rather simplified. Therefore is should not be seen as a basis for a design at this specific location. For such a design the hydraulic boundary conditions should be determined with much more accuracy. But this applies both to the “only dike” solutions as well as to the “mangrove” solution. The saving of approximately 25% of the cost will remain the same.

This part is now expanded, so that the intentions of the paper become more clear

The economic savings due to a mangrove belt are threefold:

Because of the lower wave height: The revetment can be lighter (and therefore cheaper) The crest can be lower Because of the lower crest The volume of the structure is less (and therefore cheaper) Less land has to be purchased (and thus cheaper)

In this specific example (for an industrial estate) the price of land is indeed the determining factor. It is in this case 85% of the total cost. A high price of industrial land is realistic. But even in case the value of the land is only 10% of the value assumed in the paper, the main conclusion is still valid. In percentage, the savings are even larger. The table to show this is now included in the paper.

I do not agree with your remark that this is “not be relevant in other countries where one does not have to purchase the land where coastal protection works are constructed.”. Even if the land has not to be purchased, the land has a value. For example in Vietnam where market values of (rural) land hardly exist, the land has certainly a value. For the Vietnam case (see ref 2 in the paper) therefore  not the “market” value of the land is used, but the cumulative yield of the rice produced on this land. For clarification this comment is also added to the paper.

Round 2

Reviewer 3 Report

My objection for the title is related to the fact that reading the title on expects to find in the article a study and conclusions based on several "industrial areas" with differing wave conditions but instead there is only one.

So this is a case study in my mind, and this should be shown in the title.

In the conclusions, based on this case study, one can say that the method may be applied to other sites and only postulate what will happen there.

Author Response

English has been improved in latest version. Title has been changed into "Financial benefits of mangroves for surge prone 2 high-value areas", to show that the paper is not really dealing with industrial areas, but that it focuses on areas with a high value (like industrial areas)

Round 3

Reviewer 3 Report

The new title is more appropriate.